# Cellular Plasticity in Mammary Gland Development and Breast Cancer

**DOI:** 10.3390/cancers15235605

**Published:** 2023-11-27

**Authors:** Madison N. Wicker, Kay-Uwe Wagner

**Affiliations:** Department of Oncology, Wayne State University School of Medicine and Tumor Biology Program, Barbara Ann Karmanos Cancer Institute, 4100 John R, EL01TM, Detroit, MI 48201, USA

**Keywords:** cellular plasticity, epithelial-to-mesenchymal transition, mammary gland development, tumorigenesis, breast cancer, mouse models

## Abstract

**Simple Summary:**

For many decades, the cellular and molecular mechanisms that orchestrate the differentiation of epithelial subtypes in the mammary gland have been a focus of intense investigations. Despite the orderly development of epithelial lineages with specific functions, individual cells or clusters of cells can switch identities in response to stress conditions and during the onset and progression of breast cancer. This review provides a comprehensive overview of factors that promote cellular plasticity. Changes in epithelial cell identity associated with pregnancy and lactation, inflammation, tissue repair, as well as the origin and progressive development of breast cancer subtypes discussed in this review demonstrate the broad impact of cellular plasticity on normal mammary gland development and the formation of malignant tumors.

**Abstract:**

Cellular plasticity is a phenomenon where cells adopt different identities during development and tissue homeostasis as a response to physiological and pathological conditions. This review provides a general introduction to processes by which cells change their identity as well as the current definition of cellular plasticity in the field of mammary gland biology. Following a synopsis of the evolving model of the hierarchical development of mammary epithelial cell lineages, we discuss changes in cell identity during normal mammary gland development with particular emphasis on the effect of the gestation cycle on the emergence of new cellular states. Next, we summarize known mechanisms that promote the plasticity of epithelial lineages in the normal mammary gland and highlight the importance of the microenvironment and extracellular matrix. A discourse of cellular reprogramming during the early stages of mammary tumorigenesis that follows focuses on the origin of basal-like breast cancers from luminal progenitors and oncogenic signaling networks that orchestrate diverse developmental trajectories of transforming epithelial cells. In addition to the epithelial-to-mesenchymal transition, we highlight events of cellular reprogramming during breast cancer progression in the context of intrinsic molecular subtype switching and the genesis of the claudin-low breast cancer subtype, which represents the far end of the spectrum of epithelial cell plasticity. In the final section, we will discuss recent advances in the design of genetically engineered models to gain insight into the dynamic processes that promote cellular plasticity during mammary gland development and tumorigenesis in vivo.

## 1. Cellular Plasticity

Cellular plasticity is a phenomenon where cells can change their identity during early development and tissue homeostasis in adults when they de- or transdifferentiate as part of an organ’s normal response to injury [1]. A prime example of the importance of cellular plasticity during development is the formation of the germ layers during gastrulation, which requires the epithelial-to-mesenchymal transition (EMT) of posterior epiblast cells. Recent studies like the article by Scheibner and colleagues [2] provide new insight into the mechanisms by which epithelial cell plasticity is associated with endoderm formation and how this process may differ from the genesis of the mesenchyme. The progressively complex cellular composition of tissues within developing organ systems is generally thought to occur hierarchically from stem/progenitor cells that give rise to differentiated cellular lineages. There is increasing experimental evidence that the developmental trajectory along cellular lineages is reversible (dedifferentiation) and reprogrammed mature cells acquire characteristics of tissue-resident unipotent or multipotent progenitors [3] (Figure 1A). The process by which mature cells utilize evolutionary conserved molecular programs to acquire a regenerative capacity and changes in their identity is called “paligenosis” [3,4]. While transdifferentiation can occur through a temporary progenitor-like state, selected mature cell types may deviate from their normal hierarchies and directly convert into different cell fates, often in response to inflammation or other pathological signals (Figure 1B). The phenotypic manifestation of changes in cell identity on the histological level is called metaplasia. Metaplastic lesions are mostly benign and reversible. The swift and widespread conversion of pancreatic acinar cells into duct-like cells in response to inflammation and the subsequent regeneration of acini exemplifies the importance of cellular plasticity in tissue repair. The term acinar-to-ductal metaplasia (ADM) is also used for the process by which oncogenic signals, most often mutant *KRAS*, cause the formation of precursor lesions for pancreatic ductal adenocarcinoma [5,6]. More accurately, an oncogene-driven reprogramming and change in cell fate should be defined as dysplasia where, unlike in metaplasia, the permanent activation of oncogenic signals causes preneoplastic lesions that are persistent and possess the propensity to develop into cancer. 

Cellular plasticity in normal tissue homeostasis like ADM in the pancreas generally encompasses various processes by which mature cells de- or transdifferentiate in response to environmental and genotoxic stressors [3,4]. In contrast to pancreatic development, there are notable differences in the definition of cellular plasticity in the field of mammary gland biology. Here, this process is mostly portrayed as a feature of stem and progenitor cells or a transient process by which epithelial cells acquire EMT-like characteristics such as a loss of apicobasal polarity during ductal branching morphogenesis or wound healing [7,8]. This limited perspective is likely a consequence of more than 50 years of research that primarily focused on the identification of mammary epithelial stem/progenitor cells and establishing their roles in breast cancer. In mammary tumorigenesis, cellular plasticity is most often defined as the ability of malignant cells to toggle between epithelial and mesenchymal features that may be associated with a gain in stem cell-like properties and cell motility [9]. In addition to stem cell biology and EMT, this review will highlight several other biological processes associated with cellular plasticity in normal mammary gland development and tumorigenesis. 

## 2. The Evolving Model of a Mammary Epithelial Hierarchy

More than half a century ago, work by DeOme and coworkers [10] demonstrated that any segment from the mammary epithelium from an adult donor mouse could regenerate a functional mammary gland following transplantation into an epithelial-free (i.e., cleared) mammary fat pad of a recipient female. The donor’s age and reproductive status do not have a major impact on the successful transplantation [11], but the serial passage of transplanted fragments induces a regenerative senescence [12,13]. Based on these seminal findings, research efforts focused on the identification of tissue-resident stem and progenitor cells. The mammary epithelial transplantation model became the gold standard to functionally discriminate limited numbers of heterogeneous epithelial cell populations as well as the contributions of genetically labeled individual cells in the resulting outgrowths. The studies revealed that a single epithelial cell carrying a unique integration site of the mouse mammary tumor virus (MMTV) could contribute to the formation of an entire mammary gland [14] and give rise to lobular-restricted and ductal-limited progenitors [15]. The amalgamation of these observations led to a first hierarchical model of mammary epithelial cell differentiation (Figure 2A) with ectodermal-derived stem cells and multipotent epithelial precursors as antecedents of all functionally restricted epithelial lineages [16]. A major difference between this early hierarchical model with current illustrations is the previously held notion that functionally restricted alveolar cells and ductal cells may give rise to both luminal and myoepithelial progeny as both cell types were present in the duct- or lobule-limited outgrowths. 

The identification of specific markers and their application using flow cytometry to enrich multipotent stem cells and luminal epithelial progenitors (e.g., CD24, CD29, CD49f, EpCAM/SCA1, and CD61) were significant contributions that led to a revision of the mammary epithelial hierarchy [17,18,19,20]. The updated model (Figure 2B) was more aligned with the hematopoietic system where long-term and short-term repopulating stem cells give rise to common progenitors that differentiate into luminal and myoepithelial progenitors and their mature descendants (i.e., ER+/ER− ductal cells, alveolar cells, and myoepithelial cells) [21]. The existence of myoepithelial and luminal-specific progenitors in the proliferative zone of extending mammary ducts (i.e., terminal end buds, TEBs) was also validated in transplant experiments using epithelial cells from female donors that carry a LacZ reporter on one of the two X chromosomes. The LacZ reporter is expressed in a mosaic pattern due to the random inactivation of the maternal or paternal X chromosome during embryogenesis [22]. The results of this study showed that LacZ-labeled cap cells in TEBs gave rise to the myoepithelial lineage and not luminal cells within subtending ducts. Conversely, the luminal lineage must have originated from LacZ-negative progenitors in the body cells of TEBs. These findings are in agreement with observations from Cre/lox-based genetic labeling experiments demonstrating that unipotent progenitors are the antecedents of the mature cell populations within the luminal and basal epithelial lineages during postnatal mammogenesis [23,24]. While embryonic mammary glands are initially rich in multipotent stem cells, the results from single-cell expression studies suggest that the late embryonic and most of the postanal development of the mammary gland is orchestrated by unipotent, lineage-restricted progenitors [25,26]. This does not necessarily imply that multipotent stem cells are absent in the adult gland. In a commentary, Smith and Medina [27] provide arguments for the presence of multipotent progenitors in the adult gland, and it might be premature to dismiss several studies on the presence of long-label-retaining epithelial cells and the significance of asymmetric cell division in the mammary gland, which are features of stem cells [28,29]. As part of the discussion on the developmental contributions of multipotent and unipotent stem/progenitor cells, it is correct to point out that lineage-restricted progenitors and mature cells may behave differently during normal tissue homeostasis compared to engraftment of cells into an epithelium-divested mammary fat pad [30]. The different microenvironments can instigate cellular plasticity, which can obscure lineage-restricted characteristics of epithelial subtypes as discussed in the next section. 

A discourse about the evolving model of the mammary epithelial hierarchy would not be complete without highlighting recent efforts to delineate the cellular heterogeneity during various developmental stages of the gland using single-cell genomics and proteomics. A summary of interesting observations from single-cell RNA sequencing experiments in addition to those discussed in the previous paragraph can be found in a comprehensive review by Anstine and Keri [31]. There is a wealth of new information, but the collective results of establishing the transcriptional profiles of individual cells and grouping them into distinct clusters according to similar gene expression signatures do not demand a fundamental revision of the established developmental trajectories of epithelial cell lineages with flow cytometry markers. The most important advancement from these studies, thus far, is that they demonstrate that mammogenesis from the embryonic gland throughout all major postnatal phases is a complex process where heterogeneous cell populations gradually evolve through transitional stages, including those that represent bipotent and unipotent progenitors and their differentiating descendants (Figure 2C). There is no agreement on the timing of a clear separation of the luminal and basal lineages during prenatal and early postnatal development [26,32]. The findings by Wuidart et al. [25] suggest that lineage specification occurs already during embryogenesis, confirming earlier results from immunofluorescence co-labeling studies of cytokeratins [33]. The single-cell sequencing experiments have validated the expression of genes associated with stem/progenitor cells (e.g., *Tspan8*, *Procr*, *Lgr5*) and identified new putative markers such as *Cdh5* [34]. However, the expression profiles of these genes are too varied to define a single multipotent stem cell population. Overall, the sc-RNA sequencing results are intriguing as they reveal a previously unknown heterogeneity of epithelial cell populations. Functional studies are needed to establish whether, for example, the 11 luminal and 4 basal epithelial clusters described by Bach et al. [35] or the 11 epithelial states recently reported by Kumar et al. [36] are distinct functional entities or whether they represent transient developmental stages in a mammary gland that is constantly changing in response to oscillating hormones, local growth factors, and environmental cues. It is also not clear whether selected cellular states such as the basal-luminal intermediate clusters described by Kumar et al. [36] and Gray et al. [37] are constituents of the normal epithelial hierarchy or whether they originate through cellular plasticity in the form of dedifferentiation or transdifferentiation.

**Figure 2 cancers-15-05605-f002:**
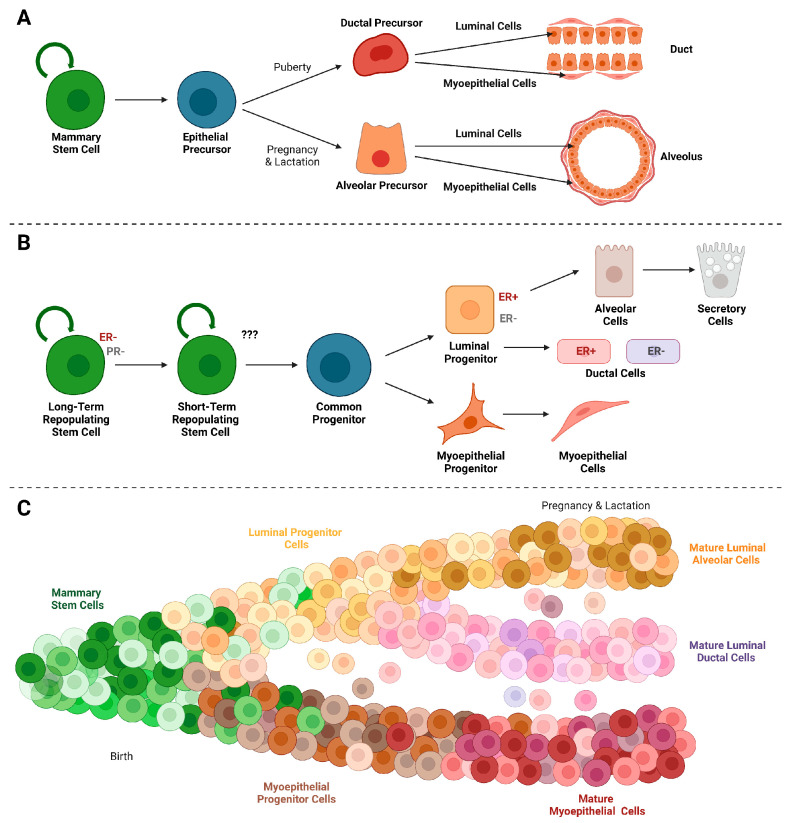
Models of the mammary epithelial hierarchy. (**A**). Differentiation of lobular-restricted and ductal-limited progenitors from pluripotent mammary epithelial stem/progenitor cells (serial transplantation model); adapted from [16] (2005). (**B**). Epithelial lineage hierarchy model based on the expression of integrins and other selected cellular markers using flow cytometry; adapted from [21]. (**C**). Continuous progression model based on transcriptomic changes and epigenetic states; adapted from [31].

## 3. Cellular Plasticity in Normal Mammary Gland Development

The model of the mammary epithelial hierarchy has become more complex and dynamic as a result of the identification of cellular states with unique or mixed gene expression profiles. The developmental trajectories of the newly identified cellular states have yet to be experimentally defined and their assigned place in the hierarchy model may not be a clear indication of whether they originated through the differentiation of progenitors or through cellular plasticity of mature cells. It is important to note that the term ‘plasticity’ in mammary gland biology is mostly used in the recent literature to describe variations in stem/progenitor differentiation along epithelial lineages, deviations in gene expression from the main cellular states, or induced alterations in cellular states in response to experimental manipulation such as transplantation or a targeted deregulated expression of genes [8,30,37]. Before the major discovery that differentiated somatic cells can be reprogrammed into a pluripotent state [38], the cloning of the first mammal from a cultured mammary epithelial cell through epigenetic reprogramming of its nucleus in an enucleated ovum is probably the most spectacular line of investigation demonstrating somatic cell plasticity [39]. Given that mammary epithelial cells can exhibit such remarkable plasticity under experimental conditions, it is surprising that de- and transdifferentiation are not commonly documented as normal processes in the mammary gland despite evidence of more extensive cellular plasticity in addition to single cells losing polarity and temporary acquiring mesenchymal features during ductal morphogenesis as mentioned earlier.

In the mid-1990s, secretory alveolar cells in the mammary gland expressing the Whey Acidic Protein (WAP) were considered ‘terminally differentiated’. WAP is a milk protein whose expression is more than 1000-fold induced by lactogenic hormones in functionally differentiated alveolar cells during late pregnancy and lactation [16,40,41]. Northern blot data comparing the transcriptional activation of milk protein genes in mice showed that *Wdnm1* and *β-casein* are expressed earlier during pregnancy, while the activation of *α-lactalbumin* and *Wap* occurs only a few days before the birth of the offspring [42]. Hence, WAP is one of the latest known markers for functional differentiation. The term ‘terminal differentiation’ was used to indicate that WAP-expressing alveolar cells were destined to die following the cessation of lactation and during the postlactational involution period. Given this notion, the development of a transgenic strain expressing the Cre recombinase under the control of the *Wap* gene promoter (WAP-Cre) to conditionally delete genes specifically in the mammary gland did not seem well justified, unless the examination of gene function is restricted to functionally differentiated cells. Indeed, the activation of the WAP-Cre transgene follows closely the expression profile of the endogenous *Wap* locus during pregnancy and lactation and both are silenced during involution. However, the examination of intercrosses of the WAP-Cre strain with a Cre/lox reporter mouse line revealed that the Cre recombinase left a permanent mark on the DNA of a significant number of epithelial cells in non-pregnant parous mice [43]. This unexpected finding suggested that not all ‘terminally differentiated’ cells expressing the late milk protein gene *Wap* undergo cell death. The availability of a Cre/lox reporter mouse line expressing LacZ under the control of the constitutively active *Rosa26* locus (*Rosa26^LSL-LacZ^*) [44] allowed the visualization of individual cells in the gland that had progressed through advanced stages of functional differentiation during lactation (i.e., *Wap* expression) but survived the involution process [45]. These adjunct epithelial cells are numerous and a permanent cell population in non-pregnant, parous females. The labeled cells are located primarily at the terminal ends of the ducts, and they no longer express Cre recombinase or endogenous WAP. Using this Cre/lox-based cell lineage tracing experiment, it was demonstrated that the LacZ-labeled cells serve as progenitors for the next generation of alveolar cells in multiparous females before they start to express WAP again [45]. We initially considered dedifferentiation as a process for the origin of these cells that we later named ‘parity-induced mammary epithelial cells’ (PI-MECs). However, following the convention of the epithelial hierarchy model of the late 1990s, which largely dismissed dedifferentiation or other forms of cellular plasticity during normal development, we proposed that PI-MECs may represent a hybrid state of a progenitor with a partial or temporary commitment to advanced differentiation. This view was enforced by the results of transplantation experiments demonstrating that LacZ-labeled PI-MECs were able to contribute to the formation of the entire ductal tree when engrafted into the cleared fat pads of wildtype recipients [45,46,47]. This suggested that PI-MECs are not just lobular-restricted progenitors as predicted from their location in the gland. Moreover, lineage tracing experiments of LacZ-labeled PI-MECs showed that these cells give rise to mammary tumors in parous MMTV-neu and MMTV-PyMT transgenic females [48,49], supporting the general notion that mammary tumors may originate from stem/progenitor cells. The Cre/lox-based cell lineage tracing experiment to identify PI-MECs was very likely not an artifact of the WAP-Cre transgene. The visualization of PI-MECs was also accomplished through pulse-chase labeling of their nuclei with the help of an H2B-GFP reporter (TetO-H2B-GFP) in mice that express the reverse tetracycline-controlled transactivator (rtTA) under the regulation of the endogenous *Wap* gene (*Wap-rtTA* knockin) [50]. *Wap-rtTA* TetO-H2B-GFP double transgenic females were treated with doxycycline (Dox) during pregnancy and early lactation to induce the expression of the H2B-GFP reporter protein that is incorporated into the nuclei of cells that express the endogenous *Wap* locus. Subsequently, Dox was withdrawn to stop the expression of the reporter in lactating females and mammary gland involution was initiated by the removal of their pups. Cells with H2B-GFP-labeled nuclei were still present at the terminal ends of ducts two and three weeks after the postlactational remodeling of the gland was complete. Unlike the Cre/lox-based labeling of PI-MECs where the reporter gene is constitutively expressed, the TetO-H2B-GFP transgene is silent due to the lack of Dox and rtTA expression, and the GFP cells are lost over time. This may suggest that PI-MECs are distinct from the small population (2%) of long-term BrdU label-retaining epithelial subtype [51]. Another interesting finding of the longitudinal analysis of multiparous *Wap-rtTA* knockin mice was that mammary glands exhibit a remarkable degree of functional plasticity and adaptation where alveolar development and the activation of the *Wap* locus are proportional to the number of lactating pups [50].

The transplantation experiment into the cleared fat pad is the gold standard to assess the potency of stem/progenitor cells, but it is certainly correct to caution the results of this methodology as it can promote cellular plasticity as discussed in recent reviews [30,31]. In support of this notion, parity-induced epithelial cells (PI-MECs) serve as alveolar progenitors in multiparous females, and they do not play any role in the formation of mammary ducts as these cells arise much later following the first gestation cycle. Only upon transplantation, this cell population contributes significantly to the formation of the entire ductal tree, including the large collecting ducts [45]. PI-MECs are not equivalent to multipotent stem cells as their total number in the involuting gland [about 16% in a mixed genetic background [47]] greatly exceeds the proposed quantity of tissue-resident multipotent stem cells [approximately 1 in 1400 according to Stingl et al. [17]]. Whether PI-MECs are considered a hybrid state of a progenitor with a partial or temporary commitment to advanced differentiation or whether they originate through dedifferentiation from a subset of fully committed lactating alveolar cells during involution, their existence in parous females is evidence for widespread cellular plasticity in the mammary gland. The occurrence of cellular reprogramming during the gestation cycle has been recently rediscovered using sc-RNA sequencing. Bach et al. [35] reported that parity induces permanent changes in the transcriptome and increases the number of luminal progenitors with an alveolar fate that may promote alveologenesis during subsequent pregnancies. We called PI-MECs the ‘functional memory’ of the parous mammary gland [52] based on experimental findings that PI-MECs are selectively amplified during subsequent gestation cycles and serve as the cellular basis for the rescue of the lactation defect in multiparous prolactin receptor-deficient mice [45]. Hence, there is experimental evidence of the proposed significance of parity-mediated changes in the transcriptomes of epithelial cells as recently determined by sc-RNA sequencing [35]. Other single-cell genomics and proteomics studies have also identified unique clusters of epithelial subpopulations with partial secretory differentiation profiles (casein expression) in the breasts of parous women [36,37]. These collective observations suggest that the lactation-involution cycle has a major impact on persistent gene expression profiles and the emergence of distinct cellular subtypes or cellular states. Additionally, there is experimental evidence of the biological role of cellular plasticity in functional adaptation. 

## 4. Cellular and Molecular Mechanisms That Promote Plasticity in the Mammary Gland

The parity-induced epithelial cells (PI-MECs) were identified in cell lineage tracing experiments that monitor the activation of the *Wap* gene promoter during pregnancy and lactation, and it is evident that these cells had to undergo a dedifferentiation process during mammary gland involution that led to the silencing of *Wap* and other milk protein genes. It is interesting to note that, similar to acinar-to-ductal metaplasia in the pancreas, PI-MECs emerge in response to inflammation which is a hallmark of postlactational remodeling. This suggests that the reprogramming of a subset of alveolar cells to become tissue-resident in parous females might be controlled by an inflammatory microenvironment. The exact mechanism for this cellular phenomenon and epigenetic reprogramming is unknown, but inflammatory cytokines and the extracellular matrix (ECM) likely play critical roles in this process. Interleukin-6 class inflammatory cytokines (LIF, OSM) are upregulated during involution and activate the JAK1/STAT3 pathway, which accelerates the death of differentiated alveolar cells [53,54,55,56]. Like ductal epithelial cells, PI-MECs within the collapsing alveoli are resistant to programmed cell death and the proapoptotic actions of inflammatory cytokines. It is interesting to speculate whether the ECM in the involuting gland promotes the reprogramming of a subset of epithelial cells. It has been reported that the composition of the ECM in the mammary gland changes during the reproductive cycle [57]. Bruno et al. [58] found that epithelial-free ECM preparations derived from involuting mammary glands can redirect the development of embryonic stem cells and suppress teratoma formation when transplanted into a cleared mammary fat pad. Earlier experiments performed in the laboratory of Gilbert Smith at the National Cancer Institute have demonstrated that the mammary gland niche is responsible for the reprogramming of cells from diverse tissue sources (brain, bone marrow, testis) toward a mammary epithelial fate [59,60,61]. An illuminating observation made by Bruno and coworkers [58] was that mammary epithelial cells are not required at all for the reprogramming process, and their extra-cellular matrix is sufficient to coax transplanted testicular cells into the development of a functional mammary gland. The collective findings from these lines of investigations suggest that the microenvironment (i.e., niche) is a critical determinant for reprogramming progenitors and possibly also more differentiated descendants, such as the precursors of PI-MECs. This notion is in line with the results of earlier experiments that demonstrated that the mesenchyme of the mammary gland or salivary gland instructs the organotypic branching patterns of their epithelium irrespective of the source of the epithelial cells from a different tissue [62,63]. 

The cell lineage tracing studies and transplantation experiments provided evidence of cellular plasticity and illuminated some important cellular determinants of this process, including the microenvironment. Specific insight into molecular mechanisms that govern cellular plasticity was gained from studies of genetically engineered models with a deregulated expression of developmental genes. NOTCH signaling is a key regulator of a variety of cellular functions and has been shown to control stem cell maintenance, cell fate decisions, and differentiation. NOTCH signaling is engaged in luminal epithelial cells and is suggested to promote the commitment of mammary stem cells toward a luminal epithelial cell fate [64,65]. A block of this signaling cascade in luminal epithelial cells of the developing gland through deletion of its main downstream effector RBP-J shows that NOTCH signaling is essential for the maintenance of the luminal cell fate [66]. Deficiency in RBP-J causes luminal cells to acquire a basal epithelial cell-characteristic expression of p63 and CK6 and co-expression of luminal and basal epithelial markers (CK14, CK18). On the phenotypic level, the results of this study may indicate that loss of NOTCH signaling causes cellular plasticity in the form of transdifferentiation (co-expression of luminal and basal keratins). Whether this process involves transient dedifferentiation of luminal cells into bipotent progenitors has yet to be determined. Experimental evidence suggests that p63, which acts antagonistically to NOTCH signaling [67], has an active role in the luminal-to-basal transdifferentiation process (Figure 3). Expression of the intracellular domain of NOTCH1 suppresses the predominant ΔNp63 isoform, which lacks the N-terminal transactivation domain and functions as a primary determinant of the basal cell fate. More relevant to the discourse of cellular plasticity, the forced expression of p63 in purified luminal cells is sufficient to confer a basal cell lineage phenotype [67]. Similar to p63, WNT signaling counteracts the biological role of NOTCH, and this antagonistic function is partially controlled by the histone methylation reader PYGOPUS 2 (PYGO2). On the mechanistic level, PYGO2 is required for the β-catenin-mediated repression of NOTCH3, and deficiency in this epigenetic regulator results in an increase in NOTCH signaling that promotes luminal epithelial differentiation [68]. Although it was reported that the overexpression of PYGO2 caused a reduction in the expression of the *NOTCH3* mRNA in MCF10A cells, the study did not examine whether a targeted upregulation of PYGO2 in differentiated luminal cells of the mammary gland is sufficient to promote reprogramming, thereby linking an epigenetic modification to the initiation of cellular plasticity. 

A recent review by Holliday et al. [69] provides a broad overview of the known epigenetic mechanisms such as DNA methylation and histone modifications that play a role in the expression of genes that govern epithelial cell lineage specificity. EZH2 is a member of the Polycomb repressor complex that catalyzes the trimethylation of H3K27 and promotes broad changes in the epigenome of mammary epithelial cells that define the luminal identity [70]. Interestingly, promoters of key developmental genes contain active and inactive histone methylation marks, keeping these bivalent regulatory elements in a transiently repressed (i.e., poised) state. In response to cell-intrinsic and external stimuli, the bivalent promoters of transcription factors controlling cell lineage determination can be swiftly activated. This might be particularly relevant for the plasticity of luminal progenitors that have intermediate promoter features between basal and mature luminal cells. As discussed later, luminal progenitors are the suggested cells of origin for a subset of basal-like breast cancers and these epigenetic mechanisms may also govern the observed luminal-to-basal transition of epithelial cells in response to oncogenic RAS [71,72]. 

While oncogenic signals such as hyperactive RAS may promote the de- or transdifferentiation of a subset of cells with luminal identity, several key transcription factors are known to restrain cellular plasticity within specific epithelial lineages. For example, the helix-loop-helix transcription factor ID4 is expressed in basal epithelial cells and suppresses luminal differentiation. Deficiency in Id4 in basal cells results in an unusual expression of the estrogen receptor and its transcriptional network in myoepithelial cells [73]. On the other hand, the Signal Transducer and Activator of Transcription 5 (STAT5) is crucial for the development and maintenance of luminal progenitors [74]. While the development of the luminal epithelial lineage is not affected by mutations or the loss-of-function of p53, the tumor suppressive and cell cycle control functions of this transcription factor limit the expansion of stem cells and luminal progenitors [75,76]. Similarly, loss of the Ets transcription factor ELF5 leads to an increase in the number of luminal progenitors, and it has been demonstrated that ELF5 is crucial for luminal lineage restriction as well as the development and functional differentiation of alveolar cells [77]. In summary, the identity of mammary epithelial lineages is sustained by epigenetic mechanisms and the functions of key transcription factors that restrain the plasticity of epithelial cells as they differentiate along their developmental trajectories. 

## 5. Cellular Plasticity during Early Stages of Mammary Tumorigenesis

Squamous metaplasia of the breast is a condition where epithelial cells become phenotypically similar to those of the epidermis (i.e., flattened appearance) and keratinized [78]. Such changes in cell identity are seen in inflamed breasts, cysts, and fibroadenoma, which is the most frequent palpable lesion in adolescents and young women. The high incidence of metaplastic changes in the normal breast emphasizes the fact that cellular plasticity is a common process during normal tissue homeostasis that is instigated by constant changes in systemic and local factors (e.g., hormonal fluctuations during the menstrual cycle, and inflammation). Experimentally, cellular reprogramming and the formation of squamous metaplasia with keratinization can be modeled in mice with chemical carcinogens and human breast tissue explant cultures by combined treatment with cyclic adenine nucleotide, prostaglandins, and papaverine [78]. Although benign metaplastic lesions are not significantly associated with a clinically meaningful increased risk of breast cancer [79], these alterations are being viewed as histologic abnormalities and consequently studied alongside breast cancer. 

Most investigations on the determinants of cellular plasticity in mammary tumorigenesis focus on the EMT process of cancer cells, as discussed in the next section, but changes in cellular identity also occur during the earliest stages of cancer initiation. Several oncogenes and tumor suppressors promote a luminal-to-basal transition of epithelial cells within preneoplasia, and this type of cellular plasticity can obscure the cellular origin of mammary tumors that present as basal-like triple-negative breast cancers (TNBCs) at the time of diagnosis. Most malignant tumors in patients that carry germline mutations in *BRCA1* are classified as basal-like TNBCs. However, the cellular and molecular analyses of tissue samples from BRCA1 carriers as well as genetically engineered mouse models deficient in BRCA1 provide evidence that TNBCs preferentially arise from luminal progenitors [80,81]. In support of this notion, the selective removal of the *Brca1* gene from K14 and K6a-positive basal epithelial cells is not sufficient to trigger the development of mammary cancer [82]. In contrast, mammary tumors form in aging mice where the *Brca1* gene is conditionally deleted using the MMTV-Cre or WAP-Cre transgenes [83]. On the mechanistic level, observations by Bai et al. [84] showed that a loss of function of the cyclin-dependent kinase inhibitor p18 (IINK4c) promotes mutant BRCA1-associated malignant transformation through an expansion of the estrogen receptor (ERα)-negative luminal progenitor pool. The subsequent work by these authors suggested that reduced expression of the GATA3 transcription factor plays a critical role in the cellular plasticity of IINK4c-deficient luminal progenitors, resulting in an accelerated formation of basal-like mammary tumors [85]. A similar downregulation of GATA3 was also observed in mice that express oncogenic KRAS and develop basal-like and claudin-low (mesenchymal-like) mammary tumors [71]. Interestingly, this process is reversible as the downregulation of oncogenic KRAS restores the expression of GATA3 and leads to a concomitant upregulation of the luminal progenitor marker CD61 (ITGB3). 

A review of the results from various recent studies suggests that cellular plasticity during mammary cancer initiation is influenced by oncogenic driver mutations as well as the cellular subtypes that are susceptible to neoplastic transformation. Unlike the expression of oncogenic KRAS, which promotes the basal transdifferentiation of alveolar progenitors (PI-MECs) and luminal cells of larger ducts [71,72], the *PIK3ca^H1047R^* gain-of-function mutation of the catalytic domain of the PI3 kinase in basal- or luminal-restricted mammary epithelial cells increases the number of luminal progenitors [86]. As a consequence of the mutant PIK3ca-mediated induction of cellular plasticity, transforming cells acquire characteristics of multipotent progenitors that contribute to tumor cell heterogeneity and mammary cancers with diverse pathological features [86,87]. Some of these variations in tumor histopathology might be driven by secondary signaling networks that are preexistent in the cells of origin. Like the opposing effects of NOTCH and WNT signaling during normal development (i.e., luminal/basal epithelial cell lineage determination, Figure 3), the persistent activation of NOTCH steers the differentiation of mutant PI3Kca-transformed epithelial cells towards a luminal fate. In contrast, the downregulation of NOTCH in association with the gain of function of PI3 kinase signaling was reported to cause squamous differentiation [88]. An interesting finding of the study by Schachter et al. [88] is that not all types of gain-of-function mutations in *Pik3ca* show similar dependencies on other signaling networks. Specifically, constitutively active NOTCH was shown to cooperate with different *Pik3ca* mutations (i.e., E545K and H1047R) during tumor initiation, but NOTCH also functioned as an allele-specific tumor suppressor in the presence of a transforming *Pik3ca^H1047R^* allele. Like NOTCH, a constitutively active mutation in β-catenin (*Ctnnb1^δex3^*) significantly shortened the tumor-free survival in female mice expressing PIK3ca^E545K^ or PIK3ca^H1047R^. However, the resulting mammary tumors exhibited squamous features that are typical for the activation of canonical WNT signaling. This suggests that the propensity of NOTCH and WNT signaling to accelerate the onset of mammary cancer is a separate entity from their divergent effects on cellular differentiation. The collective observations support the notion that changes in oncogenic signaling networks that may cooperatively promote mammary cancer initiation can instigate cellular plasticity that results in diverse developmental trajectories of transforming epithelial cells. 

## 6. Cellular Plasticity during Breast Cancer Progression

Cellular plasticity in breast cancer is most often defined as the ability of malignant epithelial cells to acquire mesenchymal and stem cell-like properties [9]. This developmental phenomenon is an important facilitator of invasion, metastasis, and therapy resistance [89]. Elucidating the molecular mechanisms that orchestrate cellular plasticity is essential for understanding the biology of cancer and the development of novel therapeutic strategies to prevent and treat metastatic disease [9,89]. A recent article by Pérez-González et al. [90] discusses intrinsic and extrinsic mechanisms that orchestrate cellular plasticity in multiple cancer types during the initiation and metastatic progression of tumors as well as the emergence of cells that develop drug resistance. For detailed insights into the cellular and molecular determinants that drive stemness and EMT, cancer cell invasion, metastasis, and colonization in distant organs along with the reinstatement of epithelial cell characteristics (i.e., mesenchymal-to-epithelial transition, MET) please refer to a recent review by Kong et al. [9]. New findings from single-cell sequencing studies provide supporting evidence that the EMT process is not a binary switch where cancer cells toggle between epithelial and mesenchymal fates. Instead, cancer cells transit through hybrid stages where they retain epithelial features and gain mesenchymal traits that modulate invasiveness and metastatic potential [91,92]. A thought-provoking finding from the analysis of pancreatic tumor models and human breast cancer cell lines is that different EMT states may facilitate divergent modes of cancer cell invasion [93]. While a collective migration of cancer cells might be more typical for those with hybrid EMT features, it was suggested that cancer cells with mostly mesenchymal-like properties preferentially invade tissues as single cells [94]. The results from elegant cell-lineage tracing experiments by Lüönd et al. [94] and Li et al. [95] may support the notion that malignant cells with more advanced EMT features have a reduced ability to invade and metastasize. These conclusions should be taken with some caution since both studies used the transgenic MMTV-PyMT line as the mammary tumor model. Female mice express the polyoma middle T (PyMT) oncogene under the control of the MMTV promoter that is active in the epithelium but not in mesenchymal cells. Therefore, cancer cells that undergo advanced EMT and assume a mesenchymal fate should lose the expression of the MMTV-PyMT transgene and die, unless they escape an MMTV-controlled expression of the oncogene or become completely independent of the oncogenic driver. This experimental limitation will be discussed further in the last section of this review. It should also be noted that except for basal-like type 2, investigations by Wang and colleagues [96] did not reveal any statistically significant differences in the survival of women with different triple-negative breast cancer subtypes, regardless of whether they are mesenchymal, basal-like 1, or luminal ER/PR/HER2 negative, which is the most closely related human TNBC subtype to the MMTV-PyMT mouse model. Similarly, patients with claudin-low TNBCs have the same dismal prognosis as those with other aggressive breast cancer subtypes (i.e., luminal B, HER2-enriched, and basal-like) [97,98], suggesting that tumors with widespread mesenchymal characteristics are not less metastatic. 

In addition to EMT, cellular plasticity plays key roles in several other phenomena associated with cancer progression. One of those occurrences is the molecular subtype switching that is repeatedly observed in metastatic breast cancers where malignant cells lose the expression or functionality of the estrogen and progesterone receptors (ER, PR) and gain expression of HER2 [99,100,101]. More interestingly, ER-positive luminal A primary tumors can give rise to metastatic lesions of the HER2-enriched subtype that is clinically presented as HER2 negative [102]. In those cases, the molecular subtype switch was reported to be associated with the gain of function of FGFR4, which is neither mutated nor amplified. The fact that genome-wide gene expression changes and molecular subtype switching are more frequently observed in metastatic tumors might indicate that changes in the tumor microenvironment facilitate the reprogramming of metastatic cancer cells.

Another phenomenon of cellular plasticity during cancer progression is the genesis of claudin-low breast cancer (CLBC). Depending on the cellular origin and developmental trajectories, CLBC could be defined as an intrinsic molecular subtype or a phenotype [97,98]. Claudin-low breast cancers represent a subset of malignancies on the far end of the spectrum of epithelial cell plasticity where cancer cells display mesenchymal properties. These cancers are equivalent to the mesenchymal (M) and mesenchymal stem-like (MSL) TNBC cases described by Lehmann et al. [103]. They exhibit elevated levels of N-cadherin and vimentin and reduced expression of genes encoding for tight junction and cell adhesion proteins (i.e., claudin 3, 4, 7 and E-cadherin) as well as luminal cell surface markers (CD24, EpCAM) [104]. On the histopathological level, many CLBCs present as metaplastic spindle-cell lesions that are distinct from tumor-infiltrating stromal cells. Since these cancers possess similarities to multipotent epithelial progenitors with regard to the expression of CD44, CD49f, and ALDH1A1, it was reasonable to propose that CLBCs originate from mammary stem cells [105,106]. In contrast to this notion, two bioinformatics studies on data sets from the Molecular Taxonomy of Breast Cancer International Consortium (METABRIC) have revealed that claudin-low breast cancers can be stratified along with other intrinsic subtypes (e.g., luminal A, luminal B, HER2-enriched) [98,107], meaning that diverse epithelial cell types may give rise to CLBC. The continuous developmental model by Fougner et al. [98] suggests that a pure claudin-low subtype emerges gradually from most intrinsic molecular subtypes (Figure 4). This model is supported by experimental studies using genetically engineered mice that sporadically develop basal-like and claudin-low primary mammary cancers [71,72]. These studies demonstrated that the two main TNBC subtypes can originate from alveolar progenitors (PI-MECs) and luminal epithelial cells of ducts that gain basal-like and mesenchymal characteristics. Cell lineage tracing studies showed that the CLBC subtype progression is not a consequence of an accumulation of cancer-associated stromal cells but a gradual EMT process of mammary tumor cells that lose the expression of luminal and later basal cytokeratins, EpCAM, and E-cadherin [71,72].

During the discourse on whether CLBCs should be recognized as a separate intrinsic breast cancer subtype or whether claudin-low should be defined as a phenotype of other subtypes, it should be noted that most of these tumors present as TNBCs at the time of diagnosis. This means that a subset of CLBCs that may have arisen from luminal A/B or HER2-enriched subtypes underwent extensive cellular reprogramming along with the acquisition of additional oncogenic drivers to compensate for the lower expression or lack of ERα and HER2. At present, unique oncogenic drivers for the CLBC molecular subtype have not been identified [108], but a common characteristic of many TNBCs, including CLBCs, are mutations in *TP53* and the associated genome instability instigated by these mutations. Interestingly, the bioinformatic analysis by Pommier et al. [107] revealed that the most recurrent feature across all CLBCs is the activation of the RAS/MAPK pathway and exceptionally high RAS signaling correlated with stem cell characteristics. While genomic alterations in the *RAS* genes are less frequent in primary breast tumors compared to other cancer types, it is interesting to note that gain-of-function mutations in *KRAS* or *HRAS* and their downstream effectors are found in TNBC cell lines that are claudin-low and most commonly used in cancer research (e.g., MDAMB-231, Hs578T, SUM159PT) [109]. Studies in genetically engineered mice demonstrated that constitutive RAS signaling is sufficient for the onset and metastatic progression of basal-like and claudin-low mammary cancer [71,72]. Interestingly, the degree of cellular plasticity and the claudin-low subtype features in these models were dependent on the persistent expression of the cancer-initiating oncogene. The upregulation of luminal progenitor markers (GATA3, CD61) in response to the suppression of oncogenic RAS signaling is validation that claudin-low mammary tumors originated from luminal epithelial cells [71]. 

## 7. Selection of Experimental Models to Study Cellular Plasticity in Tumor Initiation and Progression

A substantial wealth of information about the cellular origins of breast cancer subtypes and the molecular mechanisms that drive cellular plasticity and tumor cell heterogeneity was obtained from studies using genetically engineered mouse models. The first generation of conventional knockout mice or transgenics targeting oncogenes to the mammary epithelium with the MMTV or milk protein gene promoters (e.g., MMTV-neu, MMTV-PyMT, WAP-TAg) provided experimental evidence for the biologically significant roles of oncogenes and tumor suppressors in mammary cancer initiation. More detailed insight into the contribution of specific epithelial subtypes in mammary cancer was gained from the development of Cre/lox-based conditional knockout models and mice that express exogenous proteins in the mammary gland in a ligand-controlled manner. The latter models were instrumental in assessing the perpetual requirement of transforming oncogenes for the maintenance and progression of mammary cancers or their substitution by other oncogenic drivers (for detailed information on conditional expression systems and a list of models see previous reviews [110,111]). Advanced models combining transgenes and targeted alleles with ligand-inducible expression systems and DNA recombinases (Cre, Flp, Dre) now make it possible to express any protein or regulatory RNA in a cell type-specific manner in the gland at defined time points. This includes the temporally and spatially controlled expression of fluorescent reporters (e.g., GFP, dTomato) for cell lineage tracing experiments to monitor the EMT process of malignant mammary epithelial cells as they activate genes that encode tenascin C (*Tnc*), N-cadherin (*Cdh2*), and vimentin(*Vim*) [94,95]. 

Despite a large variety of mammary tumor models and refined expression systems for cell lineage tracing that are available today, it should be noted that most current investigations on cancer cell plasticity still employ only a few selected first-generation transgenics to initiate the formation of mammary cancer. The MMTV-PyMT line is the most frequently used model to study metastatic disease [112]. It is however important to recognize the limitations of this model and other first-generation transgenic lines in studies investigating cellular plasticity. The MMTV and promoters of milk protein genes that are being used to express oncogenes in the mammary gland are active in epithelial cells and they are silent in the stroma of the gland (i.e., fibroblast, adipocytes, and endothelial cells) (Figure 5A). The highest expression of these gene regulatory elements occurs in luminal cells, which is in line with their biological purpose (i.e., secretion of milk proteins, production, and release of MMTV into the milk). This may explain why the expression profiles of most mammary tumors in MMTV or WAP promoter-driven cancer models like the MMTV-PyMT and MMTV-neu cluster to the luminal-type and exhibit little intra-model variation [113]. Many of these tumors originate from alveolar progenitors [48,49], and recent cell lineage tracing studies confirmed that MMTV-PyMT tumors arise from cytokeratin 8 (CK8)-positive luminal epithelial cells. This includes a subset of cancer cells that gained an expression of basal cytokeratins [114]. Using tetracycline-controlled gene expression systems (i.e., MMTV-rtTA and MMTV-tTA), several studies have demonstrated that the proliferation and survival of most luminal tumor cells and descendants with partial basal-like characteristics are dependent on the sustained expression of the cancer-initiating oncogenes such as the PyMT [115], ERBB2/neu [116,117], or mutant KRAS [71,118]. Given the necessity of a perpetual expression of the driver oncogene, it should be evident that transgenes whose expression is tethered to the epithelial-specific activation of the MMTV or milk protein gene promoters are less suitable to study cellular plasticity, in particular advanced EMT that would result in the downregulation of these transgenes. Hence, the recently proposed higher propensity of hybrid-EMT cancer cells to metastasize in comparison to malignant cells undergoing more advanced mesenchymal transdifferentiation in the MMTV-PyMT model should be viewed with caution. These findings need validation in tumor models where the expression of the oncogene is sustained in a cell differentiation-*independent* manner as cancer cells progress toward an advanced mesenchymal state. 

Two main experimental approaches have been used successfully to express oncogenes in the mammary epithelium in a cell differentiation-*independent* manner (Figure 5B). The first technique is the Cre or Flp recombinase-mediated activation of oncogenic mutant forms of endogenous gene loci such as *Erbb2* [119], *Pik3ca* [86,87], and *Kras* [72]. The contribution of gain-of-function mutations of *Pik3ca* to tumor cell heterogeneity was discussed earlier. The MMTV-Flp-mediated activation of the endogenous *Kras^G12D^* in luminal epithelial cells results in tumors that cluster to the basal-like and claudin-low molecular subtypes [71]. Luminal and basal epithelial cell plasticity was also observed in tumors of mice that express the ETV6-NTRK3 fusion oncogene under the endogenous *Etv6* locus in committed alveolar progenitors expressing the WAP-Cre transgene [120]. The activation of oncogenes in the mammary gland epithelium in a cell differentiation-independent manner was also accomplished in experimental settings where the oncogene is expressed as a transgene under regulatory elements of a ubiquitously active promoter or an endogenous housekeeping gene in a Cre recombinase-dependent manner. An example is the WAP-Cre-mediated, ubiquitous expression of KRAS^G12D^ under the *Eef1a1* locus, which results in multifocal tumors with diverse histopathological features [121]. Using the elegant gene targeting approach by Klinakis and colleagues [121], Sakamoto and coworkers [122] generated mice that express the tetracycline-controlled transactivator (tTA) from the endogenous *Eef1a1* housekeeping gene (EF1) in a Cre recombinase-controlled manner [EF1-loxP-STOP-loxP-tTA] [122]. Unlike the epithelial cell-restricted expression of transgenes under the direct control of the MMTV and WAP promoter, the endogenous *Eef1a1* locus can facilitate a sustained expression of an oncogene as cancer cells transdifferentiate and undergo advanced EMT (Figure 5B). The significant impact of different modes of oncogene expression on cellular plasticity was demonstrated by comparing the molecular profiles of mammary tumors that express the same TetO-Kras^G12D^ responder transgene under the control of the MMTV-tTA or the WAP-Cre-activated EF1-tTA [71]. The MMTV-tTA-mediated expression of the TetO-Kras^G12D^ transgene resulted in luminal-type tumors with a subset of cancer cells expressing both luminal-type and basal-type cytokeratins. In contrast, the WAP-Cre-mediated activation of the TetO-Kras^G12D^ oncogene under the constitutive expression of the EF1-tTA in alveolar progenitors (PI-MECs) led to the progressive development of cells expressing both luminal-type and basal-type cytokeratins. These cells gave rise to basal-like tumors that swiftly progressed into claudin-low mammary cancers with features of advanced EMT and a high propensity to metastasize. Notably, the dissimilar cancer subtype characteristics between the MMTV-tTA and EF1-tTA transgenic models were not a consequence of differences in the expression of the mutant KRAS^G12D^ protein. The expression of mutant KRAS in tumors from both transgenic models was identical and comparable to the basal-like and claudin-low mammary tumors of mice that express the oncogene from the endogenous *Kras* locus (MMTV-Flp *FSF-Kras^G12D^*) [71]. The collective findings from the comparison of the three genetically engineered models demonstrate that an extended range of cellular plasticity is revealed in an experimental setting where cancer cells express the transforming oncogene in a cell differentiation-*independent* manner.

Despite the epithelial cell-specific dependency of oncogene expression, a subset of tumors from selected first-generation transgenic tumor models, such as the TgWAP-T121 line expressing a truncated form of the SV40 T-antigen (TAg), exhibit variations in their histopathology and gene expression profiles [113]. The occurrence of cancers with features of advanced EMT and claudin-low expression profile in the TgWAP-T121 mice is likely the result of mutations that render cancer cells independent from the tumor-initiating oncogene. In one of the first studies using the tetracycline-controlled expression system to assess oncogene addiction in cancer progression, Ewald and colleagues [123] demonstrated that advanced tumors lose their dependency on the transforming TAg oncogene. Another important element that contributes to tumor heterogeneity is the mode of action of particular oncogenes such as WNT, fibroblast growth factors (FGFs), and transforming growth factors (TGFs). These oncoproteins can act in a juxtracrine manner promoting the growth of diverse epithelial cells, which, to some extent, may obscure the plasticity of the oncogene-producing cancer cells. The cell labeling experiment by Kisseberth et al. [124] showed that tumorigenesis in mice expressing the transforming growth factor-alpha under the *Wap* gene promoter (WAP-TGFα) is preceded by the formation of polyclonal precursor lesions. The rate of polyclonality during cancer progression, however, was not investigated and similar studies are missing that demonstrate this phenomenon in cancer models that express other oncoproteins acting as ligands for juxtracrine and paracrine signaling in addition to their suggested autocrine role in cancer cells. 

## 8. Summary and Outlook

Over several decades, a major focus in the field of mammary gland biology has been the identification of mammary epithelial stem and progenitor cells as well as elucidating the hierarchical development of mammary epithelial cell lineages. The major attention on putative stem cell populations and their suggested significant roles in development and breast tumorigenesis may have contributed to a perception of cellular plasticity that is limited to stem cell function or EMT. In comparison to other exocrine organs like the pancreas where cellular plasticity plays pivotal roles in tissue regeneration, the terms dedifferentiation and transdifferentiation are infrequently used to describe developmental phenomena in the biology of the mammary gland. In this review, we highlighted changes in cell identity and the emergence of cellular states associated with the gestation cycle that are associated with dedifferentiation. We also discussed the effects of the microenvironment (i.e., the niche) on cellular reprogramming and molecular pathways that control lineage determination that, when deregulated, can instigate the transdifferentiation of epithelial cells. Several of these pathways play key roles in mammary tumor formation where they promote diverse developmental trajectories of transforming epithelial cells. There is increasing experimental evidence that luminal progenitors and the luminal-to-basal transdifferentiation of their descendants contribute to the formation of basal-like mammary cancers. During tumor progression, EMT is mostly portrayed as a temporary cellular state that promotes invasion and metastasis. The more widespread transdifferentiation of luminal and basal tumor cells towards a mesenchymal fate and the manifestation of the claudin-low breast cancer subtype is a prime example of cellular plasticity. It should be noted that human TNBC cell lines that are commonly used in biomedical research, including MDA-MB-231 cells, are claudin-low [97]. This suggests that most studies on TNBC have been conducted on cellular models with extensive changes in cellular identity (more than 20,000 studies on MDA-MB-231 cells alone according to PubMed). Recent progress in the generation of genetically engineered mouse models that sporadically develop metastatic basal-like and claudin-low triple-negative mammary tumors provides opportunities to study the cellular and molecular mechanisms that drive luminal-to-basal and mesenchymal transdifferentiation. Using these new models, it is important to recognize that they mediate the expression of the tumor-causing oncogene in a cell differentiation-independent manner. This mode of oncogene expression can be expected to reveal the full extent of plasticity that cellular subtypes assume in response to the expression of a particular oncogene. This is not the case in most conventional transgenic breast cancer models where the oncogene is tethered to an epithelial-specific promoter. Moreover, if the neoplastic characteristics and changes in cancer cell identity are being persistently upheld by the oncogenic driver, as demonstrated for RAS signaling in the recent study by Rädler et al. [71], models similar to the one shown in Figure 5B (exogenous expression) can be used to study the reversibility of cancer cell de- and transdifferentiation in vitro and in vivo. This includes investigations of the plasticity of cancer cells that remain dormant following a ligand-mediated downregulation of the oncogenic driver.

## Figures and Tables

**Figure 1 cancers-15-05605-f001:**
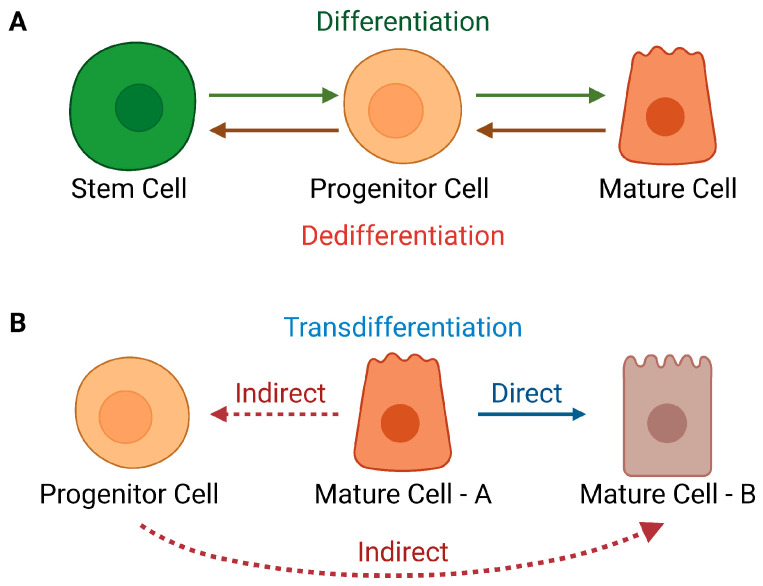
Changes in the identity of mature cells that portray plasticity in lineage hierarchies. (**A**). Dedifferentiation (**B**). Transdifferentiation, direct or indirect, via transient progression through a progenitor state; adapted from [4].

**Figure 3 cancers-15-05605-f003:**
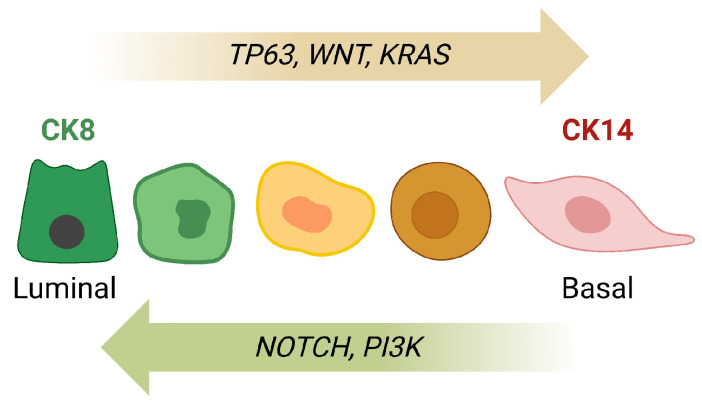
Molecular drivers and signaling pathways that orchestrate epithelial cell lineage determination and cellular plasticity. Cytokeratin 8 (CK8)-positive luminal epithelial cells can acquire basal cell characteristics through expression of TP63, WNT, and KRAS. Conversely, epithelial cells expressing basal cytokeratins (e.g., CK14) can transdifferentiate into luminal cells in response to NOTCH and PI3K signaling.

**Figure 4 cancers-15-05605-f004:**
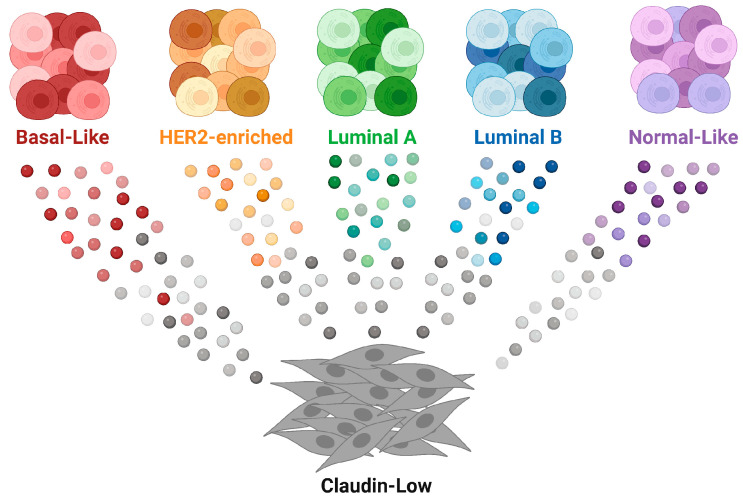
Continuous model of the development of mesenchymal-like claudin-low breast cancer. Tumors acquire varying degrees of claudin-low characteristics and progress into a distinct claudin-low tumor type that is uncoupled from other intrinsic molecular subtypes; adapted from [98]. According to this model, the claudin-low breast cancer subtype is the cellular and molecular manifestation of cellular plasticity.

**Figure 5 cancers-15-05605-f005:**
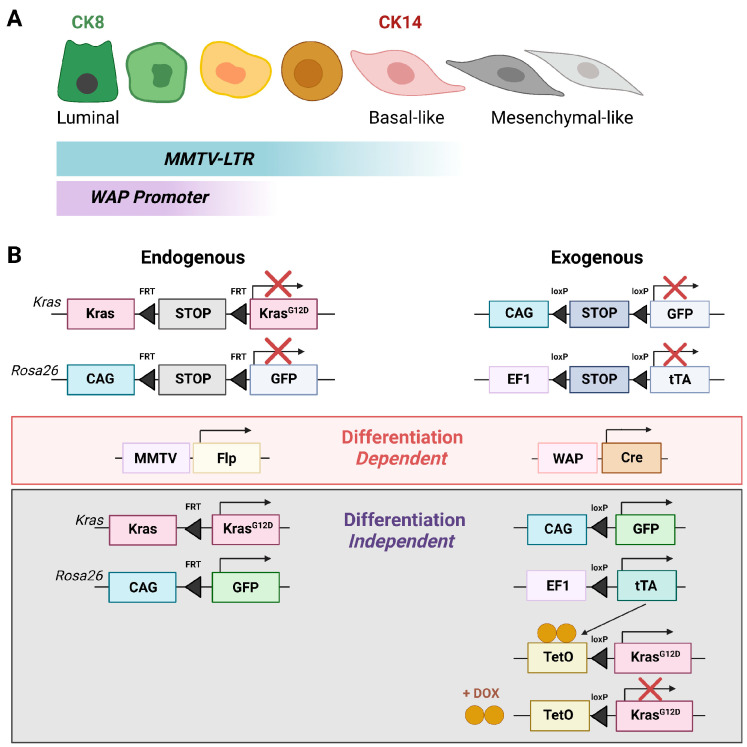
Advances in the development of experimental models to study cellular plasticity in tumor initiation and progression. (**A**). Illustration of the epithelial cell lineage-dependent activation of the MMTV-LTR and *Wap* gene promoter that are used to target the expression of oncogenes to the mammary gland. Note that the activation of these regulatory elements is highest in the luminal epithelium and absent in the mammary stroma. Therefore, the expression of the oncogene is tethered to cells that retain epithelial characteristics as long as the cancer cells are dependent on the oncogene for their growth and survival. (**B**). Two experimental approaches to express oncogenes in the mammary epithelium in a cell differentiation-*independent* manner. In these models, the mammary epithelial cell lineage-dependent activation of oncogenes (or reporter genes for cell lineage tracing) is mediated by the transient expression of DNA recombinases (Cre, Flp) that activate oncogenic mutant alleles of an endogenous gene (left) or the expression of transgenes that activate oncogenes from a ubiquitously active promoter or gene locus (right). An advantage of the transgenic (exogenous) expression model is its versatility in that oncogenes can be expressed in a ligand-mediated (i.e., doxycycline-controlled) as well as differentiation-independent manner to study the roles of oncogenes in the maintenance of cellular plasticity (e.g., mesenchymal properties of the claudin-low breast cancer type).

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
