# Peer review of "Cellular Plasticity in Mammary Gland Development and Breast Cancer"

_cancers, 2023, doi:10.3390/cancers15235605_

Round 1

Reviewer 1 Report

Comments and Suggestions for Authors

This is a very well written, comprehensive review looking at mammary plasticity. The only things that could have improved this review was the inclusion of information regrading epigenetic mechanisms that influence cell plasticity but this information may be covered elsewhere or been beyond the scope of this review. 

Author Response

Thank you for your kind and thoughtful comments. Following the discussion of the histone methylation reader PYGOPUS 2 as a molecular determinant that promotes plasticity in the mammary gland, we added a paragraph on epigenetic mechanisms that control mammary epithelial cell lineage determination. We cite an excellent review by Holliday et al on this subject. and highlight the role of EZH2 as well as active and inactive histone methylation marks on bivalent promoters of developmental genes that may contribute to cellular plasticity.

Reviewer 2 Report

Comments and Suggestions for Authors

This is a well-written and comprehensive review of mammary gland development and breast cancer.  It was a pleasure to read this summary of transgenic models from the author’s research work on the plasticity of mouse mammary gland remodeling.  It included sc-RNA seq data and relevant questions for future analysis.  I have a few minor suggestions for improvement.

Specific comments:

1.    Inconsistent for detailed description of genes/protein names, e.g., pg 4- paragraph under Fig. 2 legend uses std. abbrev., but on pg. 7, section 4, all the terms are defined at length.

2.    Pg. 9 second paragraph in section 5- you already abbreviated EMT and pg. 11- MMTV was likewise previously abbreviated. You might wish check this type of issue throughout the text w/ a quick search and edit.

3.    Pg. 13- top line: you could include the names of the TNBC cell lines from 99 (so the reader doesn’t have to go look it up).

Author Response

We are glad this reviewer found the manuscript informative, in particular the discussion of genetic models.

Specific comment 1: Following the suggested changes, the text only uses standard gene or protein names, and consequently the sentences are easier to read.

Specific comment 2: Both issues were corrected.

Specific comment 3: As suggested by the reviewer, we have listed 3 of the most commonly used claudin-low TNBC lines that have both RAS and effector mutations, e.g. RAF, PI3K. 

Reviewer 3 Report

Comments and Suggestions for Authors

This well written review does a very good job of documenting mammary cell plasticity in various experimental models and pathological conditions. From the text, it seems that over the past few decades, the concepts of mammary stem and progenitor cells have become more muddled, subject to constraints imposed by experimental systems and markers employed, rather than any overarching biological justification. Despite the manuscript’s already voluminous length, I would have liked to see some discussion of what is known about the factors that ordinarily restrain cellular plasticity. Are these commitment factors primarily intrinsic or extrinsic? Under what circumstances are niche factors dominant and how do oncogenic drivers overcome this dominance? Answering these important questions would transform this manuscript from a competent review to a major accomplishment.

Author Response

We thank the reviewer for the positive response and several thought-provoking comments. In addition to the discussion of epigenetic mechanisms that orchestrate cell lineage determination, we also now highlight the roles of several key transcription factors (ID4, p53, STAT5, ELF5) that maintain lineage identity and restrict the growth of progenitors, thereby restraining cellular plasticity. Extrinsic factors, the role of the ECM, and how the niche controls the development of mammary epithelial cell lineages are discussed in another section of the manuscript. We added two classic papers that describe how the mesenchyme dictates the growth of the epithelium from other glands or tissues in an organ-specific manner (salivary and mammary glands). Determining the mechanisms by which oncogenes overcome lineage restriction and promote plasticity is the subject of current investigations by our team.